# Effect of Viscosity on the Formation of Porous Polydimethylsiloxane for Wearable Device Applications

**DOI:** 10.3390/molecules26051471

**Published:** 2021-03-08

**Authors:** Dong-Hyun Baek, Hachul Jung, Jeong Hun Kim, Young Wook Park, Dae Wook Kim, Ho Seob Kim, Seungjoon Ahn, Young-Jin Kim

**Affiliations:** 1Department of Display/Semiconductor Engineering, College of Engineering, Sunmoon University, Asan-si 31406, Korea; dhbaek@sunmoon.ac.kr (D.-H.B.); zerook@sunmoon.ac.kr (Y.W.P.); phykdw@sunmoon.ac.kr (D.W.K.); hskim3@sunmoon.ac.kr (H.S.K.); sjan@sunmoon.ac.kr (S.A.); 2Center for Next-Generation Semiconductor Technology, Sunmoon University, Asan-si 31406, Korea; 3Integrated Medical Technology Team, Department of Research & Development, Medical Device Development Center, Osong Medical Innovation Foundation, Cheongju-si 28160, Korea; hacher99@kbiohealth.kr (H.J.); jeonghun@kbiohealth.kr (J.H.K.)

**Keywords:** porous structure, stretchable electrode, polydimethylsiloxane (PDMS), wearable device

## Abstract

Medical devices, which enhance the quality of life, have experienced a gradual increase in demand. Various research groups have attempted to incorporate soft materials such as skin into wearable devices. We developed a stretchable substrate with high elasticity by forming a porous structure on polydimethylsiloxane (PDMS). To optimize the porous structure, we propose a manufacturing process that utilizes a high-pressure steam with different viscosities (400, 800, 2100, and 3000 cP) of an uncured PDMS solution. The proposed method simplifies the manufacturing of porous structures and is cost-effective compared to other technologies. Porous structures of various viscosities were formed, and their electrical and mechanical properties evaluated. Porous PDMS (3000 cP) was formed in a sponge-like three-dimensional porous structure, compared to PDMS formed by other viscosities. The elongation of porous PDMS (3000 cP) was increased by up to 30%, and the relative resistance changed to less than 1000 times with the maximum strain test. The relative resistance increased the initial resistance (R_0_) by approximately 10 times during the 1500-times repeated cycling tests with 30% strain. As a result, patch-type wearable devices based on soft materials can provide an innovative platform that can connect with the human skin for robotics applications and for continuous health monitoring.

## 1. Introduction

In recent years, medical devices have been developed for various diagnostic purposes to maintain the quality of life and enhance public health. The effective development of medical services has given rise to an aging society. The demand for medical devices and wearable health care devices is rapidly increasing with the rise in the global aging population [1], and such devices are expected to greatly contribute to the reduction of medical expenses. The convergence and development of information and communication technology (ICT) and bio-micro-electromechanical systems (bio-MEMS) technology have accelerated the development of wearable devices that can monitor vital signs continuously, diagnose diseases, and provide treatment.

Several research groups have developed skin-attached wearable devices as multi-arrayed electrode for recording vial signals using parylene-C [2], polyimide (PI) [3,4,5], and polydimethylsiloxane (PDMS) as the substrate and encapsulation material. Furthermore, PDMS has been extensively used in bio-chips for cell culture because of its excellent biocompatibility and gas permeability [6]. Although PDMS has excellent formability because of its low surface energy, the development of method is challenging to keep the adhesion between the metal layer and PDMS [7]. Eventually, metal deposition and patterning method on PDMS has received attention because PDMS has good flexibility, stretchability, and similar Young’s modulus suitable for skin curvature attachment. Several methods have been developed to tackle the challenge, such as injection of metal ions onto the PDMS surface [8], mixing conductor materials with polymers [9], and composite carbon nanotubes (CNTs) with polymers [10,11,12,13]. Moreover, others have reported using inorganic materials coated with a soft materials such as hydrogels [14].

Human skin has a relatively low Young’s modulus compared to skin-attached wearable devices. In other words, when a slight external force is applied between flexible substrate such as PDMS and the formed metal layer, the layer is delaminated effect to mechanical mismatch such as cracks or wrinkles. To address these problems, it is necessary to develop a skin-attached wearable device with flexible material such as PDMS, which have mechanical properties similar to human skin [15,16,17]. The limitations in the mechanical properties of inorganic materials used in conventional wearable devices must be addressed by developing alternatives. An essential requirement for the development of skin-attached wearable devices is a low modulus of elasticity, such as that of the skin, for conformal contact on the skin and effective biocompatibility.

Figure 1a represents the Young’s modulus as a color map, and it shows that each material has specific intrinsic values for it. The physical properties change depending on the internal structure of PDMS in Figure 1b, and the porous PDMS is generally more resilient to stress than the non-porous PDMSs. 

In this paper, we propose a simple and cost-effective method to realize a porous structure, which can be used as a stretchable and flexible substrate, under steam at high pressure. We compared the pore size and penetration depth of the porous structure at varying viscosities of the uncured PDMS solution. Subsequently, we evaluated the mechanical properties of the porous PDMS substrates made of different viscosities through the maximum strain and cycling strain test. The substrates were tested as wearable devices, and are expected to be suitable for wearable devices, such as skin patch platforms and other wearable electronic applications.

## 2. Results 

### 2.1. Mechanical Properties

We measured the viscosity of uncured PDMS solutions mixed at ratios of 1:1 wt% (400 ± 1.5 cP), 2:1 wt% (800 ± 1.5 cP), 5:1 wt% (2100 ± 1.0 cP), and 10:1 wt% (3000 ± 1.5 cP). All samples were thermally cured under identical conditions in the porous structure formation process. The mechanism of the porous PDMS formation was achieved under high-pressure steam passed through the uncured PDMS surface. The high-pressure steam in the chamber transferred sufficient energy to permeate the top surface of the uncured PDMS solution. This enabled the permeated steam to form cavities of various dimensions depending on the penetration depth. 

Figure 2 shows the flat layer of all samples was similar, about ~100 μm thickness under the porous layer, and the thickness of the porous layer was related to the viscosity. The pore size varied depending on the depth of steam penetration; the sizes of the pores reduced the depth from the surface increased. Therefore, the thickness of the formed porous layer was determined based on the size of the permeated steam. The pore sizes ranged from 3 to 20 μm, and the penetration depth was intense depending on the high viscosity, and the porous layer thickness increased, as shown in Figure 2.

Figure 3a–d shows top-view FE-SEM images of the porous layers with 1:1, 2:1, 5:1, and 10:1 wt% uncured PDMS, respectively. Overall, the pore size was uniform under all viscosities. However, the pore density varied according to the uncured PDMS viscosity. In 10:1 wt%, the high-pressure steam deeply penetrated from the surface of the uncured PDMS solution, thereby resulting in a lot of pores. It was concluded that the uncured PDMS viscosity affected the formation of various porosity densities.

Figure 4 shows the surface of the porous PDMS with 10:1 wt% subjected to varying strain conditions. Figure 4a,b show that they slightly elongate corresponding to the strain direction under 10% and 20% strains. The 30% strained porous structure grew denser with reducing surrounding structure distance, and stretched in the strain direction resulting in breakage, as shown in Figure 4c. In contrast, the porous structure, and the deposited metal layer of the 40% strained porous PDMS, were disrupted and their electrically contact broken in Figure 4d.

### 2.2. Electric Stability Test

The changes in the relative resistance (ΔR/R_0_, ΔR indicate the changed resistance during the strain test; R_0_ indicates the initial resistance) evaluated the value of the one-way strain and the maximum stretching ability by various viscosity (Figure 5). 

There, contact was broken at approximately 18% and 22% strain in 1:1 wt% and 2:1 wt% conditions, respectively. On the contrary, the relative resistance increased 10 times for 5:1 wt% compared to the initial resistance at 12% strain and over 100 times at 15% strain. The relative resistance gradually increased till 15% strain; however, it increased rapidly after reaching 30% strain. In 10:1 wt% condition, the relative resistance increased to approximately 10 times the initial resistance at 14% strain and was increased by 100 and 1000 times at 22% and 30% strain, respectively. The relative resistance of the low viscosity group showed a dramatic change in the relative resistance at a strain lower than that of the high viscosity group. As shown in Figure 2 and Figure 3, the density of the porous layer was low, and the depth of the porous layer was low under the conditions of 1:1 and 2:1 wt%. Conversely, in high viscosity groups (5:1 and 10:1 wt%), a sponge-like structure was formed with a higher porous density and thicker porous depth than those of the low viscosity groups.

To evaluate the electric stability of the porous PDMS, we measured the relative resistance during a 1500-cycling test in a uniaxial direction up to 30% strain (2%/s rate) using 5:1 wt% (red line) and 10:1 wt% (orange line) samples (Figure 6). As the cycling tests progressed, the deposited metal layer (gold) suffered physical damage because of the maximum strain, resulting in the continuous increase in the relative resistance. The relative resistance for 10:1 wt% resulted in the smallest relative resistance fluctuation, which increased by approximately 10 times during the cycling tensile test. As shown in the magnified graph of the 300, 600, 900, and 1200 cycles in Figure 6, the relative resistance appeared to slightly increase as the number of cycles increased. Furthermore, while 10:1 wt% recovered to the initial state, the relative resistance was identical to that of the initial value. However, the 5:1 wt% solution exhibited a phenomenon, where the relative resistance was lower than the initial resistance. This is caused by the rearranged porous structure formed by the ductility property of the deposited metal layer through repeated deformations.

## 3. Discussion

In previous studies to develop stretchable substrate, UV ozone treatment could form the wavy structure that provided good and stretching performance of PDMS substrate [18]. Sugar was used as a template to form the highly elastic 3D porous sponge-like PDMS sponge [19]. The following is a study applied as a conductor in a flexible and stretchable substrate. Conductive filler or particles embedded in an elastomer matrix [20], filling the liquid alloy metal in PDMS [21]. The PDMS has the advantage of excellent biocompatibility and stretchability, but it has several disadvantages. Several studies have been conducted to solve the problems. We developed a biocompatible and stretchable PDMS substrate using a simple and cost-effective method, which prevented the more widespread applications to biomedical fields. This proposed method can be formed to realize a high density of pores and large percentage of empty space to achieve the high porous layer elongation of up to 30%. The corresponding relative resistance changed the initial relative resistance approximately 100 times. Finally, these results demonstrated that the stretchable substrates can be applied as wearable devices capable of monitoring bio-signals such as electrocardiogram (ECG), electromyogram (EMG), blood pressure, and artificial skin electronics. Therefore, it is expected that the porous PDMS can be applied to stretchable research fields such as skin patch platforms and wearable electronic applications.

## 4. Materials and Methods

### 4.1. Porous PDMS Fabrication 

The schematic describes the mechanism of a porous structure formation with high-pressure steam in Figure 7. We prepared an uncured PDMS solution (Sylgard^®^184, Dow corning Co., Seoul, Korea) by varying the PDMS base, and curing agent ratios were 10:1, 5:1, 2:1, and 1:1 by weight% (wt%). 

To produce the uniform thickness PDMS film with different viscosity, a 100-µm thick PI tape (5413K, 3M) was attached to the edge of the carrier glass substrate (5 × 7 cm), and each viscosity uncured PDMS solution was poured along the attached polyimide tape on the glass substrate to prevent overflow and flattened with slide glass to prepare the sample. The uncured PDMS solution on the carrier substrate was placed inside a homemade high-pressure steam chamber to expose the uncured PDMS solution to pressurized steam with 1000 mL of deionized (DI) water. The uncured PDMS solution was cured for 10 min using the 15 psi of steam from the boiling water in the chamber. The pressurized steam was provided by boiling water to produce the surface of the uncured PDMS solution into a porous structure. After the process, the sample was dried in a convection oven at 70 °C for 10 min to evaporate the trapped moisture in the porous PDMS surface. A metal layer consisting of 200 nm of gold (Au) and 20 nm of titanium (Ti) was deposited on the porous PDMS using an electron-beam evaporator (SNTEK) to evaluate the mechanical and electrical properties.

### 4.2. Mechanical and Electrical Characterization 

We compared the viscosity PDMS base to curing agent mixing ratio at 10:1, 5:1, 2:1, and 1:1 wt% with a viscometer (IKA ROTAVISC lo-vi advanced, Seoul, Korea). Each 6.7 mL of uncured PDMS solution was administered in the viscometer chamber that is volume limited spindle set (VOL-SP-6.7, IKA ROTAVISC) with 2.0% viscosity accuracy and 0.2% viscosity repeatability, and each viscosity was measured for each condition with an auto-rpm (range: 0.1~200 rpm) setting for 10 min. The surface of the porous PDMS made in various ratios was coated with gold with a thickness of 10 nm, and the top and sides were observed with field emission scanning electron microscopy (FE-SEM) (JSM-7100, HITACHI). A low acceleration beam voltage of 5 kV was used to minimize polymer damage. The pore size was determined as the average value measured in the 1 × 1 cm range of five points by the obtained SEM images (both top-view and side-view).

While the porous PDMS was strained to its maximum length, we observed the surface of the porous PDMS. In addition, the test samples were deposited with Au (200 nm) and Ti (20 nm) on a porous PDMS, whose size were 6 cm in length and 1 cm in width with rectangular shape. They were fixed at stationary shaft, then the other end of Porous PDMS was attached of the x-axis stage (PG430, SURUGA SEIKI CO, Tokyo, Japan) at distance of 2 cm. The electrical properties of porous PDMS fabricated four different viscosities were evaluated at a maximum tensile strain of 30% and 1500-cycle tensile strain test in Figure 8, and the x-axis strain stage stretched the test sample at a rate of 2% per second and simultaneously measured the resistance change with five-digit multimeter (34401A, Agilent).

## 5. Conclusions

We verified experimentally that the high-pressure steam facilitated the formation of the porous structure, and we examined the influence of viscosity on the formation of the porous structure including the penetration pore depth and pore density. The high-pressure steam played a significant role in forming a porous layer, which supplied sufficient energy that allowed the water vapor to penetrate the uncured PDMS solution to form a porous layer. In the depth direction of the porous layer, the penetration depth of the steam varied depending on the mixed curing agent. However, the pore size of the porous structure was relatively uniform, ranging from 3 to 20 μm. Furthermore, the empty space formed by the penetrated steam increased the thickness of the porous structure. As a result of the mechanical evaluation, it was found that the mechanical properties of the porous PDMS were improved compared to the flat PDMS (conventional PDMS curing method) because of the pore thickness and pore density. The porous PDMS comprises a larger percentage of empty space, like a sponge, than the flat PDMS, and the porous layer induced an enhanced elasticity of PDMS. In the low-viscosity group, the porous layer was formed with a thin and low pore density, and the resistance increased rapidly before gradually increasing at 10% strain. In contrast, in the 10:1 wt% condition, in which the formed porous layer was thick, a stretching electrical property was maintained for up to 35% strain. The cycling tensile strain test results showed that the relative resistance of 10:1 wt%, which has a sponge structure, exhibited the smallest fluctuation. Finally, these results indicated that the optimized viscosity to fabricate a stretchable substrate was 10:1 wt%.

## Figures and Tables

**Figure 1 molecules-26-01471-f001:**
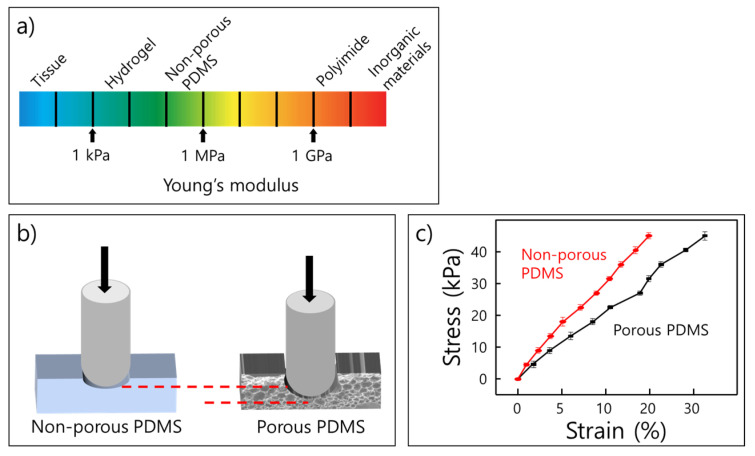
(**a**) Color map presents Young’s modulus of various materials such as hydrogel, polymer, and inorganic materials [14]; (**b**) schematic indicating that the elasticity is modified by the formed porous structure; (**c**) graph shows stress–strain curve of non-porous and porous PDMS.

**Figure 2 molecules-26-01471-f002:**
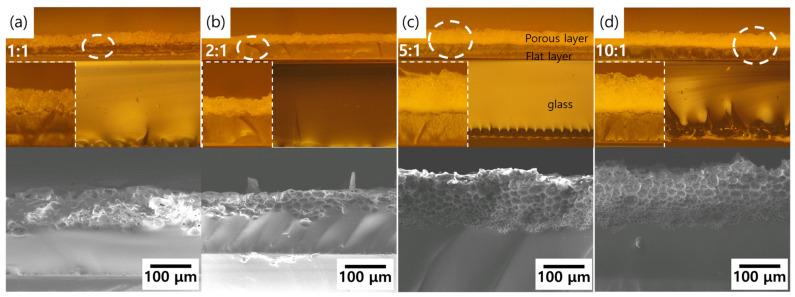
Optical and SEM images showing porous layer formed by high-pressure steam process with (**a**) 1:1 wt% (400 cp); (**b**) 2:1 wt% (800 cp); (**c**) 5:1 wt% (2100 cp); and (**d**) 10:1 wt% (3000 cp) uncured PDMS solution.

**Figure 3 molecules-26-01471-f003:**
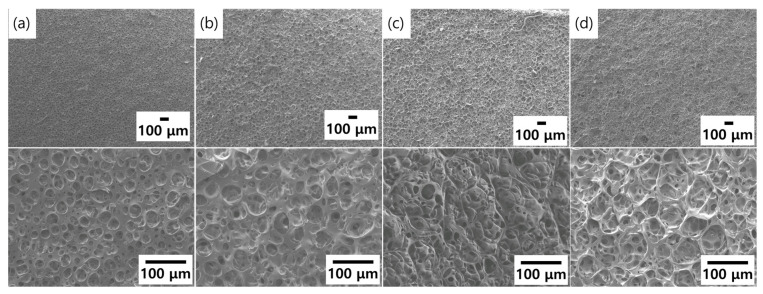
FE-SEM images of surface of the formed porous layer with (**a**) 1:1 wt% (400 cp); (**b**) 2:1 wt% (800 cp); (**c**) 5:1 wt% (2100 cp); and (**d**) 10:1 wt% (3000 cp) of uncured PDMS solution.

**Figure 4 molecules-26-01471-f004:**
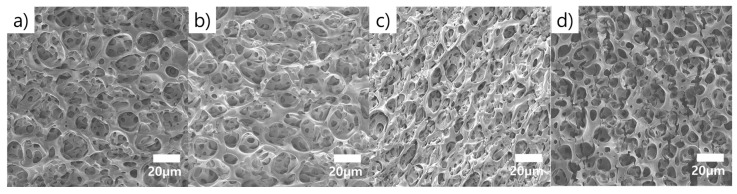
FE-SEM images showing metal (Ti/Au) deposited onto the porous PDMS surface during application of tensile strain with (**a**) 10%; (**b**) 20%; (**c**) 30%; and (**d**) 40% strain.

**Figure 5 molecules-26-01471-f005:**
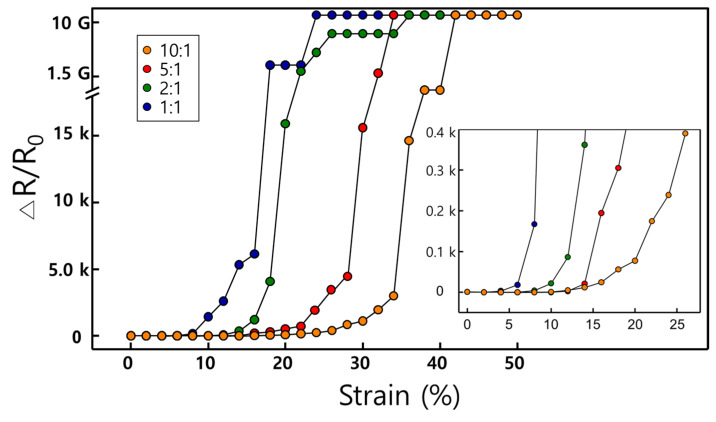
The graph shows the relative resistance (ΔR/R_0_) with the maximum tensile strain test using porous PDMS fabricated with various viscosities. Inset graph shows magnify relative resistance until 25% strain.

**Figure 6 molecules-26-01471-f006:**
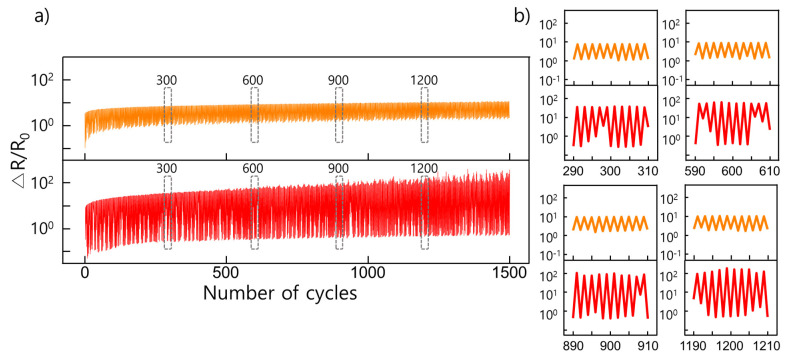
(**a**) Plots showing cycling strain test results over 1500 cycles with 30% strain; (**b**) Plots are magnified images of the 300th, 600th, 900th, and 1200th cycles in (**a**); the change in the resistance for each number of cycle is shown.

**Figure 7 molecules-26-01471-f007:**
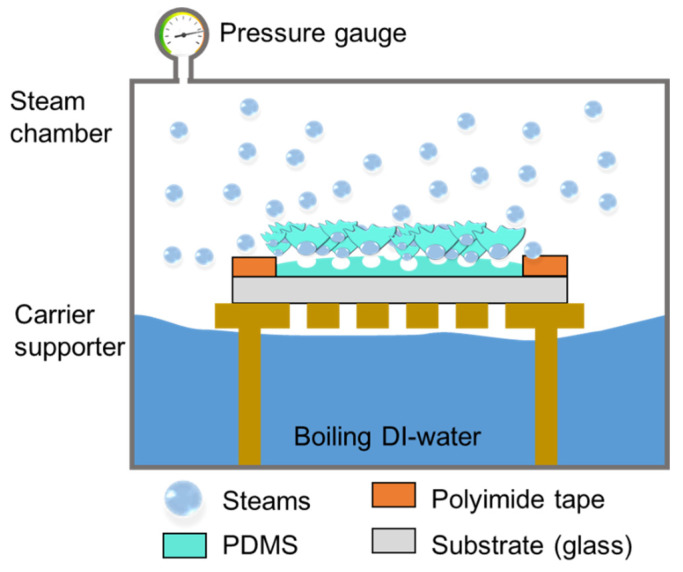
Schematic of mechanism of the formation of porous structure from uncured PDMS solution through high-pressure vapor process.

**Figure 8 molecules-26-01471-f008:**
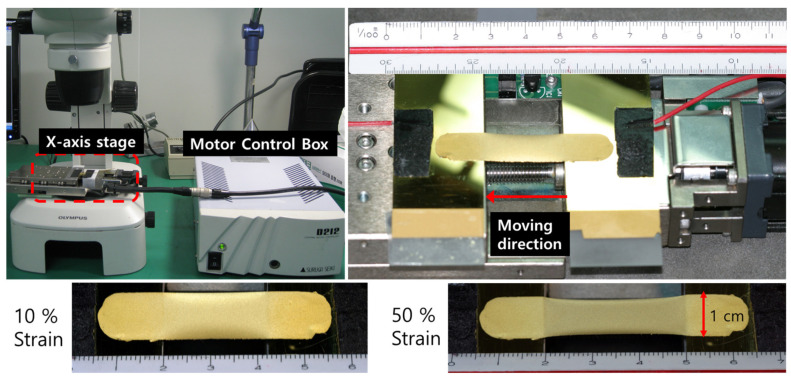
Uniaxial tensile strain and cycling strain test setup.

## Data Availability

Not applicable.

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
