# Peer review of "Effect of Viscosity on the Formation of Porous Polydimethylsiloxane for Wearable Device Applications"

_molecules, 2021, doi:10.3390/molecules26051471_

Round 1

Reviewer 1 Report

This is an interesting study with promising results. I have only  few comments:

The description of the high-pressure vapor process could be a little bit more elaborate, e.g. how exactly the substrate is being fixed into position in the high-pressure chamber and so on. The corresponding image is a bit stylized. Which electron-beam evaporator was used to deposit the coatings?

English language is okay, except for a few typos which should be eliminated by careful reading (e.g. "non-taxic" in line 212).

Author Response

We appreciate reviewer's informative comments, and we thoroughly read the comments and answered item by item faithfully as possible. We highlighted the revised sentence in manuscript.

Reviewer 2 Report

The authors investigated the rheological properties of the formation of porous polydimethylsiloxane to be used on development of medical wearable devices. Materials showing different porous structures, and their electrical and mechanical characteristics were evaluated. The authors concluded that the optimized soft material can be used for developing innovative platform to connect the human skin for robotics applications and for continuous health monitoring.

The manuscript subject is very interesting; however, some parts of it need to be correct and explained:

Q1. Introduction section is poor and it should be rewritten, explaining better the interface skin and material, and also between metal layer and substrate.

Q2. Is the Fig. 1 new or from any reference? Moreover, where the graphic of letter C is from?

Q3. Experimental procedures should be better explained so they can be reproduced by other researchers. In this sense, it is necessary to include more details in methodologies.

Q4. What about the description of materials?

Q5. Line 82 - What was the pressure of water vapour?

Q6. Lines 93-96 - This is discussion and not methodology.

Q7. Line 97 - What was the geometry/spindle of the viscometer?

Q8. Lines 99-100 - It is necessary to explain with more details the FE-SEM and pore size analyses.

Q8. Lines 103-109 - Explain better these methods.

Q9. Line 129 - How was determined the size range?

Q10. Lines 211-222 - The discussion is after the conclusion. Moreover, the discussion is very poor and not based on previous studies/articles. The authors should improve it!

Q11. A low number of references were considered.

Author Response

(The authors gave the same response as above.)

Round 2

Reviewer 2 Report

Most of the questions were addressed by the authors. However, the manuscript shows some problems yet. The authors should consider the journal's style and format. Please, verify the author guidelines. Moreover, a deep revision should be performed regards problems of English language grammar and typos.